# Leptin Is an Important Endocrine Player That Directly Activates Gonadotropic Cells in Teleost Fish, Chub Mackerel

**DOI:** 10.3390/cells10123505

**Published:** 2021-12-11

**Authors:** Hirofumi Ohga, Kosuke Ito, Kohei Kakino, Hiroaki Mon, Takahiro Kusakabe, Jae Man Lee, Michiya Matsuyama

**Affiliations:** 1Aqua-Bioresource Innovation Center (ABRIC) Karatsu Satellite, Kyushu University, Saga 847-0132, Japan; 2Laboratory of Marine Biology, Faculty of Agriculture, Kyushu University, Fukuoka 819-0395, Japan; k.ito@agr.kyushu-u.ac.jp; 3Laboratory of Insect Genome Science, Faculty of Agriculture, Kyushu University, Fukuoka 819-0395, Japan; k.kakino@agr.kyushu-u.ac.jp (K.K.); mhiro@agr.kyushu-u.ac.jp (H.M.); kusakabe@agr.kyushu-u.ac.jp (T.K.); 4Laboratory of Creative Science for Insect Industries, Faculty of Agriculture, Kyushu University, Fukuoka 819-0395, Japan; jaemanle@agr.kyushu-u.ac.jp; 5ABRIC, Kyushu University, Fukuoka 819-0395, Japan; rinya_m@agr.kyushu-u.ac.jp

**Keywords:** chub mackerel leptin, silkworm pupae, follicle-stimulating hormone, luteinizing hormone

## Abstract

Leptin, secreted by adipocytes, directly influences the onset of puberty in mammals. Our previous study showed that leptin stimulation could promote the secretion of follicle-stimulating hormone (FSH) and luteinizing hormone (LH) from pituitary cells in primary culture and ovarian development in chub mackerel. This study aimed to elucidate the detailed mechanism of leptin-induced effects on gonadotropin hormone-producing cells. We produced recombinant leptin using silkworm pupae and investigated the effects of leptin on FSH and LH secretion and gene expression in the primary culture of pituitary cells from chub mackerel. The presence or absence of co-expression of *lepr* mRNA, FSH and LH b-subunit mRNA in gonadotropic cells was examined by double-labeled in situ hybridization. The addition of leptin significantly increased the secretion and gene expression of FSH and LH from male and female pituitary cells in primary culture. In contrast, gonadotropin-releasing hormone 1 affected neither FSH secretion in cells from females nor *fshb* and *lhb* expression in cells from males and females. The expression of *lepr* was observed in FSH- and LH-producing cells of both males and females. The results indicate that leptin directly regulates gonadotropin synthesis and secretion and plays an important role in the induction of puberty in teleost fish.

## 1. Introduction

Leptin is a 16 kDa hormone secreted by fat cells [1]. It mediates its signal through the membrane receptor LepR, which belongs to the class I cytokine receptor family [2]. Adequate nutrition is required for the initiation of puberty, and leptin plays an important role in initial maturation by transmitting the amount of energy stored in the body to the reproductive axis [3]. In fact, there is a high correlation between circulating leptin and gonadotropin (GTH) levels during the pubertal process [4]. Administration of leptin to the periphery can induce puberty efficiently [5,6]. Malnutrition reduces circulating leptin levels and suppresses sexual function; however, the condition can be rescued by leptin administration [7,8].

In mammals, the pulsed secretion of GTH-releasing hormone (GnRH) activates GTH secretion and initiates pubertal development; however, GnRH neurons are known to have no leptin receptor. Recent studies suggested the possibility of an indirect mechanism of action [9]. LepR is expressed in gonadotrophs of the pituitary and is directly involved in the regulation of follicle-stimulating hormone (FSH) and luteinizing hormone (LH) secretion [10,11,12]. In studies using rats, leptin stimulation was found to efficiently promote the secretion and production of FSH and LH from primary cultured pituitary cells [13,14,15,16,17]. In particular, the pituitary is highly responsive to physiological concentrations of leptin from a young age [17]. The action of leptin on the pituitary plays an important role, independent of GnRH, and is an essential signal for puberty.

Prior to the discovery of fish leptin, studies with European sea bass (*Dicentrarchus labrax*) and rainbow trout (*Oncorhynchus mykiss*) reported that heterologous mammalian leptin stimulates LH secretion from primary cultured pituitary cells [18,19]. In 2005, the first fish leptin was identified in tiger puffer [20]. Piscine leptin in teleosts is mainly produced in the liver, and despite similarities in the structures of mammalian and fish leptins, the amino acid sequences are poorly conserved. Moreover, mouse leptin has been shown not to bind to fish LepR [21,22].

In 2020, we produced recombinant leptin in the chub mackerel (*Scomber japonicus*), a marine teleost fish, and investigated its effect on reproduction-related factors both in vitro and in vivo. The addition of leptin to primary cultured pituitary cells effectively promoted the secretion of both FSH and LH [22]. In addition, repeated intramuscular administration of leptin promoted ovarian maturation in pre-pubertal females [22]. Thus, we were the first to report that leptin is involved in the regulation of GTH secretion in fish and plays an important role in fish puberty.

In our previous study, the mechanism of leptin-induced effects in gonadotropic cells was unclear. This study aimed to investigate the effects of leptin on GTH secretion and synthesis and its mechanism of action in detail. The recombinant leptin used in the experiment was produced using silkworm (*Bombyx mori*) pupae, unlike the one produced in the *Escherichia coli* expression system in the past. This was performed to overcome the inefficient production of recombinant leptin in the previous expression system. The activity of leptin expressed in silkworm pupae was compared to that of leptin expressed in *E. coli*, and the expressed leptin was added to the pituitary cells in the primary culture system to verify its effect on the production and secretion of FSH and LH. To elucidate the mechanism of action of leptin in the pituitary, the presence or absence of expression of *lepr* mRNA in FSH- and LH-producing cells was examined by double-labeled in situ hybridization.

We succeeded in producing chub mackerel recombinant leptin in silkworm pupae. The recombinant leptin produced increased FSH and LH secretion and gene expression in both males and females in vitro. In addition, the expression of *lepr* in FSH- and LH-producing cells of the pituitary revealed, for the first time in fish, that the regulation of FSH and LH secretion and production is directly controlled by leptin.

## 2. Materials and Methods

### 2.1. Production and Purification of Recombinant Leptin

The open reading frame (ORF) of chub mackerel leptin-A (amino acids 1–161 aa, GenBank: KP635449) was cloned into the modified pFastBac vector, pFastBac L21-GG-cTEVdH8STREP, which contains the *ccdB* cassette flanked by two *Bsa*I sites with unique overhangs for the Golden Gate cloning system [23]. It also allows the expression of target proteins with two his-tags, a Strep-tag, and a TEV protease cleavage site at the C-terminus (Figure 1a). Since the cm-lepA gene has a *Bsa*I site, primers were designed to remove the *BsaI* site without altering the amino acid sequence. Equimolar amounts of two amplified PCR products and pFastBac L21-GG-cTEVdH8STREP were mixed with *Bsa*I (New England Biolabs, Ipswich, MA, USA) and T4 DNA ligase (New England Biolabs). The reaction was performed for 15 cycles, each of 5 min at 37 °C and 10 min at 16 °C. Thereafter, 1 μL of the reaction mixture was transformed into *E. coli* DH10B.

C-tagged cm-lepA recombinant baculoviruses were created using *B. mori* nucleopolyhedrovirus (BmNPV) Qd04 strain [24] bacmid DNA, as described previously [25]. The bacmid DNA was then transfected into BmN (Funakoshi, Tokyo, Japan) cells using Avalanche-Everyday (EZ Biosystems, College Park, MD, USA) to generate recombinant virus particles. The culture supernatant was harvested following centrifugation at 100× *g* for 10 min at 4 °C, as the P1 virus, on the 4th day after cell transfection. The stock of high-titer virus (P3) was prepared after serial infection of cultured cells with baculovirus.

In order to express C-tagged recombinant leptin in silkworm pupae, the recombinant viruses (1 × 10^5^ plaque-forming units per pupa) were injected into silkworm pupae (day 1) of the n17 strain (provided by the Institute of Genetic Resources, Kyushu University Graduate School). On the 4th day after inoculation, the pupae were stored at −80 °C until further use.

For the purification of C-tagged leptin, a two-step purification protocol was performed based on the presence of C-terminal His8-tag and Strep-tag, as described previously [26]. Briefly, for His-tagged chromatography, 10 pupae were suspended in ice-cold extraction buffer (20 mM Tris-HCl pH 7.4, 0.5 M NaCl, cOmplete Protease Inhibitor Cocktail (1 tablet/50 mL, Roche), and 20 mM 1-phenyl-2-thiourea) and homogenized. The mixture was centrifuged at 52,500× *g* for 30 min at 4 °C to remove large debris and lipids. The supernatant was filtered using a 0.45-μm filter (Merck, Burlington, MA, USA), and crude extracts were loaded onto a 5-mL His-Trap excel column (Cytiva, Marlborough, MA, USA). His-tagged leptin was eluted with the elution buffer (20 mM Tris-HCl pH 7.5, and 0.5 M NaCl) containing 100 mM and 500 mM imidazole and confirmed by 12% SDS-PAGE. Subsequently, fractions containing His-tagged leptin were collected and concentrated by ultrafiltration using Amicon Ultra-15 3K filters (Merck), diluted in phosphate-buffered saline (PBS) (−) buffer, and further applied to a column of 5 mL Strep-Tactin Superflow (IBA Lifesciences, Göttingen, Germany) for Strep-tag chromatography. The Strep-tagged leptin was eluted with a PBS (−) buffer containing 2.5 mM desthiobiotin. Eluted fractions were further concentrated using Amicon 3K filters (Merck). After dialysis with PBS (−), the final yield of leptin was determined using YabGelImage software (https://sites.google.com/site/yabgel/, accessed on 1 December 2021); bovine serum albumin (BSA) was used as a standard. The purified protein was stored at −80 °C until further use.

### 2.2. Luciferase Reporter Gene Assay

Chinese hamster ovary (CHO) cells were suspended in Ham’s F12 Nutrient Mixture (Thermo Fisher Scientific, Waltham, MA, USA) containing 10% fetal bovine serum (Biowest, Nuayer, France), 1% of the HT supplement (Thermo Fisher Scientific), 100 U/mL penicillin, and 100 μg/mL streptomycin mixture (Nacalai Tesque, Kyoto, Japan) and cultured in a humidified incubator at 37 °C and 5% CO2. The *lepr* open reading frame sequence of the chub mackerel (GenBank: KP635451) was digested with the restriction enzymes *Nhe*I (NIPPON Genetics, Tokyo, Japan) and *BamH*I (NIPPON Genetics) and then subcloned into a pcDNA3.1 (+) expression vector (Thermo Fisher Scientific). Before reaching confluence, the cells were seeded into a 6-well plate at 6 × 10^5^ cells/well. After 24 h, the expression vector (1 μg/well), pSTAT3-Luc (1 μg/well; Signosis, Santa Clara, CA, USA), and pRL-TK (8 ng/well; Promega, Madison, WI, USA) were transfected using the X-treme GENE HP DNA Transfection Reagent (Roche Diagnostics, Basel, Switzerland). The cells were harvested at 24 h after transfection and plated in 96-well plates at 3 × 10^4^ cells/well. After 24 h, the culture medium was changed to a serum-free medium containing silkworm pupae or *E. coli*-produced leptin (10^−12^ to 10^−7^ M) and then incubated for 3 h. The luciferase activity in the cell lysate was measured using the Dual-Luciferase^®^ Reporter Assay System (Promega).

### 2.3. In Vitro Leptin Bioassay

Chub mackerel was hatched from fertilized eggs at Aqua-Bioresource Innovation Center Karatsu satellite and cultured in a 50-ton outdoor tank with a natural photo period and natural water temperature. Pituitaries were collected from pre-pubertal females and males and washed thrice with Dulbecco’s PBS (D-PBS) (−) (Nacalai Tesque) supplemented with 100 U/mL penicillin, 100 μg/mL streptomycin, and 0.25 μg/mL fungizone. Next, the pituitaries were gently shaken in D-PBS containing 0.1% collagenase type I (Sigma-Aldrich, St. Louis, MO, USA) for 3 h at 20 °C. After enzymatic treatment, the cells were dispersed by pipetting, filtered through a nylon mesh cell strainer (40 μm), and centrifuged at 200× *g* for 10 min for recovery. The cells were washed with D-PBS (−), centrifuged again, and resuspended in L-15 medium (pH 7.4, Thermo Fisher Scientific) containing 10% fetal bovine serum (Biowest), 25 mM HEPES (Thermo Fisher Scientific), 100 U/mL penicillin, and 100 μg/mL streptomycin (Nacalai Tesque). Viable cells were quantified under a microscope using the trypan blue dye exclusion method. The cells (50,000 in 250 μL) were plated into each well of a 96-well plate and cultured in L-15 medium at 20 °C. After 4 days of culture, the wells were washed with serum-free L-15 medium and incubated for 1 h in serum-free medium containing leptin (1 to 100 nM) or 10 nM of chub mackerel GnRH1 synthetic peptide (pyro-QHWSYGLSPG-NH_2_) (*n* = 10). The bioactive gonadotropin levels in the culture medium were estimated using CHO cells transfected with chub mackerel LH-R and FSH-R as described previously [27]. The cells were stored at −80 °C until analysis of leptin-stimulated gonadotropin gene expression. At the time of sampling, the fish was carefully treated and sacrificed according to the Kyushu University, Japan’s guidelines for animal experiments.

### 2.4. RNA Preparation and Quantitative Real-Time PCR

ISOGEN II (Nippon Gene, Tokyo, Japan) was used to extract the total RNA from the cells. The extracted RNA was used to synthesize cDNA using the PrimeScript™ RT reagent kit with gDNA Eraser (Perfect Real Time) (Takara, Shiga, Japan). All steps were performed according to the manufacturer’s protocol. Quantitative real-time PCR analysis was performed using an Mx 3000P quantitative PCR system (Stratagene, La Jolla, CA, USA). The FSH and LH b-subunit (*fshb* and *lhb*) transcripts were quantified using a standard curve. Ribosomal protein L8 (*rpl8*), a housekeeping gene that is stably expressed in the pituitary of chub mackerel, was also quantified using a calibration curve. The PCR mixture (10 μL) contained 1 μL of the sample or standard cDNA, 0.1 μM primer sets, 3.75 μL of PCR-grade water, 0.05 μL of ROX dye, and 5 μL of Brilliant III Ultra-Fast SYBR Green QPCR master mix (Agilent Technologies, Santa Clara, CA, USA) with the reactions performed in duplicate. For the negative control, we used 1 µL of water as the template and confirmed that there was no non-specific amplification in all cases. The PCR conditions were as follows: 95 °C for 5 min, followed by 40 cycles of 95 °C for 10 s and 60 °C for 30 s. Melting points for all primer pairs ranged between 59 °C and 60 °C. The list of primers is presented in Table 1.

### 2.5. Dual-Label In Situ Hybridization

To determine whether FSH- and LH-producing cells co-expressed *lepr* mRNA, we performed double-label in situ hybridization using a mixture of fluorescein isothiocyanate (FITC)-labeled *fshb* and *lhb* and digoxigenin (DIG)-labeled *lepr* probes. The complete open reading frames (348 and 444 for chub mackerel *fshb* (GenBank: JF495132) and *lhb* (GenBank: JF495133), respectively) were amplified by PCR. To detect *lepr* (open reading frame: 3450), we amplified a 784-bp sequence by PCR. The primer sets are shown in Table 2. These PCR amplification products were inserted into the pGEM-T easy vector (Promega) and linearized with restriction enzymes. Sense and antisense RNA probes were transcribed using FITC- or DIG-labeling mix (Roche Diagnostics) and SP6 or T7 RNA polymerase (Roche Diagnostics) according to the manufacturer’s protocol. The synthesized RNA probes were purified using NucleoSpin^®^ RNA Clean-up XS (Takara).

Pituitary cells collected from pre-pubertal females and males were dispersed by digestion using collagenase as described above and then washed thrice with D-PBS (−); furthermore, 10,000 cells/10 μL were spotted onto MAS-coated glass slides (Matsunami Glass, Osaka, Japan) and air-dried at 37 °C. In addition, the female-derived pituitaries were flash-frozen in the Tissue-Tek OCT compound (Sakura Finetech, Tokyo, Japan) using liquid nitrogen. The pituitary tissue was sectioned coronally at 8 μm thickness using a cryostat (Cryo Star NX70, Thermo Fisher Scientific) at −15 °C and mounted onto MAS-coated glass slides (Matsunami Glass). The cells and sections were post-fixed in 4% paraformaldehyde (Nacalai Tesque) for 15 min. Double-label in situ hybridization was performed using a tyramide signal amplification (TSA) plus fluorescence system (PerkinElmer, Waltham, MA, USA) and TSA plus cyanine 3 (Cy3) system (PerkinElmer) according to the manufacturer’s protocol. For hybridization, the probe concentrations were 500 ng/mL for *fshb* and *lhb* and 2000 ng/mL for *lepr* when using the frozen sections and 50 ng/mL for *fshb* and *lhb* and 100 ng/mL for *lepr* when using isolated cells. Hybridization was performed with RNA probes dissolved in a solution containing 50% formamide (Nacalai Tesque), 2× saline-sodium citrate, 5× Denhardt’s solution (Nacalai Tesque), and 40 μg/mL Baker’s yeast RNA (Sigma-Aldrich) for 18 h at 58 °C. To introduce peroxidase, peroxidase (POD)-conjugated anti-FITC antibody (diluted by 250-fold with blocking buffer; PerkinElmer and POD-conjugated anti-DIG antibody (diluted by 1000-fold with blocking buffer; Roche Diagnostics) were used. Finally, the sections were immersed in 2 μg/mL DAPI solution (Dojindo, Kumamoto, Japan) for 5 min. The reaction was stopped by washing the sections with running water and immediately covering them with CC/Mount (Diagnostic BioSystems, Pleasanton, CA, USA). The fluorescence and differential interference contrast (DIC) images were observed under an LSM-700 confocal laser-scanning microscope (Carl Zeiss, Jena, Germany) with a plan APOCHROMAT 10×/0.45 and 63×/1.4 Oil DIC lens (Carl Zeiss). Images were processed with the ZEN 2012 Black edition (Carl Zeiss). In all cases, unlabeled cells were detected in the negative-control sections using the sense probe.

### 2.6. Statistical Analysis

GraphPad Prism 8 (GraphPad Software, San Diego, CA, USA) was used for all graph creation and data analyses. Data are represented as the mean ± SEM and were analyzed by one-way ANOVA, followed by a Dunnett’s multiple comparison test (vs. control group).

## 3. Results

### 3.1. Production of Recombinant Leptin-a

The current study was designed to express the soluble form of leptin (aa 22–161). In order to express endotoxin-free leptin protein, we employed the silkworm–baculovirus expression vector system (silkworm–BEVS). As shown in Figure 1a, we constructed the expression vectors for C-terminal tagged leptin. The three affinity tags included 8xHis, Strep, and 6xHis (namely, dH8STREP) and were applied to consecutive affinity chromatography.

We used Nickel and Strep-Tactin affinity chromatography to capture leptin proteins from the hemolymph of silkworm pupae (Figure 1b). As seen in lanes CE-FT (left panel) and EA-FT (right panel) of Figure 1b, leptin proteins are effectively bound to the His-Trap and Strep-Tactin columns, respectively. The expression level of eluted C-tagged leptin (approximately 20.21 kDa including tags) in the two-step purification was apparent upon SDS-PAGE, demonstrating that the amount and purity of the protein obtained were reasonable. As determined by the BSA assay, approximately 0.1 mg of leptin (0.1 mg/10 pupa) protein was obtained.

Leptin expressed in silkworm pupae showed dose-dependent binding to the receptor, similar to that expressed in *E. coli*, as reported by Ohga et al., 2020 (Figure 2a). At 100 nM, silkworm pupa-expressed leptin showed the same bioactivity as *E. coli*-expressed leptin (Figure 2b).

### 3.2. Effect of Recombinant Leptin on GTH Secretion

Recombinant leptin expressed in silkworm pupae was added to primary cultured pituitary cells, collected from pre-pubertal fish, and the effect on FSH and LH secretion was investigated. In females, leptin stimulated FSH secretion in a dose-dependent manner and FSH secretion was significantly increased (122%; *p* < 0.05 vs. control) upon addition of 100 nM leptin (Figure 3a). LH secretion was also significantly promoted by addition of 1 nM (119%; *p* < 0.01 vs. control) and 100 nM (123%; *p* < 0.001 vs. control) leptin (Figure 3b). GnRH1 also stimulated LH secretion (116%; *p* < 0.05 vs. control) but not FSH secretion (Figure 3a,b). In males, FSH secretion was significantly promoted by addition of 1 nM (116%; *p* < 0.05 vs. control) and 100 nM (123%; *p* < 0.001 vs. control) leptin (Figure 3c). Moreover, LH secretion was significantly promoted by addition of 1 nM (119%; *p* < 0.01 vs. control) and 10 nM (114%; *p* < 0.05 vs. control) leptin (Figure 3d). GnRH1 stimulated both FSH (117%; *p* < 0.05 vs. control) and LH (122%; *p* < 0.001 vs. control) secretion in males (Figure 3c,d).

### 3.3. Effect of Leptin on GTH Gene Expression

The gene expression levels of *fshb* and *lhb* in leptin-treated cultured cells were quantified by real-time PCR. In females, *fshb* expression was significantly increased (2-fold; *p* < 0.01 vs. control) in cells treated with 10 nM leptin (Figure 4a). Similarly, the *lhb* expression level was significantly increased (10-fold; *p* < 0.0001 vs. control) in the 1 nM leptin-treated group (Figure 4b). In males, the expression level of *fshb* showed an increase in the 1 nM leptin-treated group, although the difference was not significant (Figure 4c). The *lhb* expression level showed a significant increase (1.8-fold; *p* < 0.05 vs. control) in the 10 nM leptin-treated group (Figure 4d). In both sexes, GnRH1 did not affect the expression levels of either *fshb* or *lhb* (Figure 4).

### 3.4. Expression of lepr in GTH-Producing Cells

The presence of *lepr* expression in FSH- and LH-producing cells was examined using pituitary cells spotted on a glass slide. Expression of *lepr* was observed in FSH- (Figure 5a,c) and LH-producing cells (Figure 5b,d) in both females (Figure 5a,b) and males (Figure 5c,d).

### 3.5. Localization Analysis of lepr Using Frozen Sections

The distribution of FSH- and LH-producing cells and *lepr* in the anterior pituitary was examined using frozen sections prepared from the pituitary of females. Several FSH-producing cells were seen to co-express *lepr* (Figure 6a–f). Co-expression of *lepr* was also observed in the majority of the LH-producing cells (Figure 7a–f).

## 4. Discussion

In this study, we successfully produced recombinant leptin using silkworm pupae, and the recombinant leptin showed the same receptor-binding ability as that of the leptin produced in *E. coli*. In our previous studies, recombinant leptin was synthesized using an *E. coli* expression system. However, most of the expressed protein formed inclusion bodies. The inefficiency and cost associated with refolding were the challenges faced in the application of *E. coli*-produced leptin. A recombinant protein expression system using eukaryotes rather than prokaryotes can more efficiently produce functional recombinant proteins. Furthermore, it can be used to produce proteins that require posttranslational modification (although leptin has no special modifications). In particular, an expression system using insects does not require special equipment for handling cultured cells and can be easily scaled up by increasing the number of insects. The silkworm has a history of being bred for sericulture in Japan since ancient times and is the most suitable insect for the production of recombinant proteins. In the future, various in vivo treatments will be performed using recombinant leptin produced in silkworm pupae, and the details of the effect of leptin administration on the development of gonads in chub mackerel will be investigated.

Leptin stimulation significantly increased the secretion of FSH and LH from primary cultured pituitary cells in both males and females compared to that of the control. This is compatible with previous results [22], wherein leptin was prepared from *E. coli*, and the promotion of FSH and LH secretion by leptin can therefore be considered a highly reproducible and reliable mechanism. In another study [28], which examined leptin function in males, leptin stimulation did not promote LH secretion in male primary cultured pituitary cells. In this study, pituitaries were sampled from male fish just before puberty, whereas in the previous study, fish from a more juvenile stage was used, owing to which the basal LH content in the pituitary may have been extremely low. In fact, in male salmonids, circulating LH levels were shown to increase in the late phase of sexual maturation [29,30,31]. In addition, our previous studies had shown *lhb* expression in the pituitary of male chub mackerel to be low in the early stages of sexual maturation [32]. In mammals, studies in rats reported leptin to be directly involved in the regulation of FSH and LH secretion from the pituitary gland [13,14,15,16,17]. This function may be common from teleosts to mammals; therefore, it would be interesting to investigate the effects of leptin on GTH secretion using various fish species.

In addition, we compared the effect of GnRH1 on GTH secretion in vitro. In Perciformes, a later-evolved group of teleost fish, three types of GnRH were reported [33]. Our previous study showed that GnRH1 neurons are localized in the pre-optic area of the chub mackerel brain and that GnRH1 is a hypophysiotropic GnRH form [34,35]. However, our previous in vitro study showed that GnRH1 only triggered LH secretion and had no effect on FSH secretion in female chub mackerel [27]. This result was reproduced in an experiment with the female pituitary cells in this study. Similar findings were also reported in red seabream (*Pagrus major*) [36]. Furthermore, *gnrh1*-knock out medaka (*Oryzias latipes*) showed normal ovarian development, suggesting no direct relationship between GnRH1 and FSH [37]. From these facts, GnRH may be widely accepted not to be involved in the regulation of FSH secretion in teleost fish, especially females. In this study, and in those reported previously [22,28], leptin was shown to increase FSH secretion from both female and male primary cultured pituitary cells. Our previous studies showed that purified chub mackerel FSH promotes vitellogenesis through estradiol-17β secretion [38,39]. In addition, *fshb* expression in the pituitary gradually increased from the early phase of puberty in females and males [32]. In other words, leptin is one of the important regulators of FSH secretion and may play a role in the success of puberty by activating vitellogenesis and spermatogenesis through the promotion of FSH secretion. It was noteworthy that GnRH1 stimulated FSH secretion in male pituitary cells. Our recent studies showed that intramuscular administration of GnRH1 in pre-pubertal males increases *fshb* expression in the pituitary [28]. There may be sex differences in the relationship between GnRH and FSH, which would be one of the important themes to be investigated in the future.

The present study further showed that leptin stimulation increased *fshb* and *lhb* expression in primary cultured pituitary cells. Similar results were reported in common carp (*Cyprinus carpio*), in which the addition of recombinant carp leptin to pituitary cells upregulated *fshb* and *lhb* expression within an hour [40]. In rats, the addition of leptin to primary pituitary cells increased the intracellular FSH and LH contents [15,17]. Leptin not only stimulated the secretion of GTH but also promoted its production, and this function is likely to be common to mammals and teleost fish. Of note, in the present study, the magnitude of male *fshb* and *lhb* expression fluctuated due to low leptin stimulation than in females. In addition, though the FSH secretion was promoted in males, there was no significant increase in *fshb* expression. The difference between females and males in their responsiveness to leptin has not yet been explained. In the future, quantification of *lepr* expression in the pituitaries of females and males and the variation in expression in different developmental stages of the gonads need to be investigated. On the other hand, GnRH1 did not affect the expression levels of *fshb* and *lhb* in females and males. Intramuscular injection of GnRH analogs was reported to increase *lhb* expression in the pituitary of male chub mackerel [41]. Similarly, in sockeye salmon (*Oncorhynchus nerka*) and European sea bass (*Dicentrarchus labrax*), administration of GnRHa increased the expression level of *lhb* in the pituitary [42,43]. However, the effect on GTH gene expression in vitro remains to be clarified.

Finally, we investigated the expression of *lepr* mRNA in each of the FSH- and LH-producing cells spotted on glass slides to elucidate the mechanism of action of leptin in GTH secretion and production. Both FSH- and LH-producing cells were shown to express *lepr* mRNA in both males and females. Therefore, leptin secreted from the liver was suggested to act directly on GTH-producing cells in the pituitary to promote the secretion and production of FSH and LH. Analysis combining ISH and immunohistochemistry reported that *lepr* is expressed in LH-producing cells in common carp; however, FSH-producing cells were not investigated in the study [40]. This is the first report of leptin directly targeting FSH-producing cells in fish. Furthermore, analysis using pituitary sections confirmed that *lepr* mRNA was expressed widely and co-expressed with several FSH- and LH-producing cells. Although this study did not investigate the pituitary site in detail, FSH- and LH-producing cells are known to be specifically distributed in the proximal pars distalis in chub mackerel [44]. The distribution of *lepr* was investigated by ISH in European sea bass and was shown to be expressed in the pituitary [45]. In our previous study, RT-PCR revealed *lepr* expression to be higher in the pituitary gland than in the brain of chub mackerel [46,47]. Similar expression patterns were also reported in tiger puffer (*Takifugu rubripes*) [48], tilapia (*Oreochromis niloticus*) [21], and European sea bass [45]. Given these facts, the pituitary could be a major target for leptin in fish; leptin is an important player that directly activates FSH- and LH-producing cells, leading to successful puberty.

In conclusion, we succeeded in producing chub mackerel recombinant leptin in silkworm pupae. The recombinant leptin produced increased FSH and LH secretion and gene expression in both males and females in vitro. In addition, the expression of *lepr* in FSH- and LH-producing cells of the pituitary revealed, for the first time in fish, that the regulation of FSH and LH secretion and production is directly controlled by leptin.

## Figures and Tables

**Figure 1 cells-10-03505-f001:**
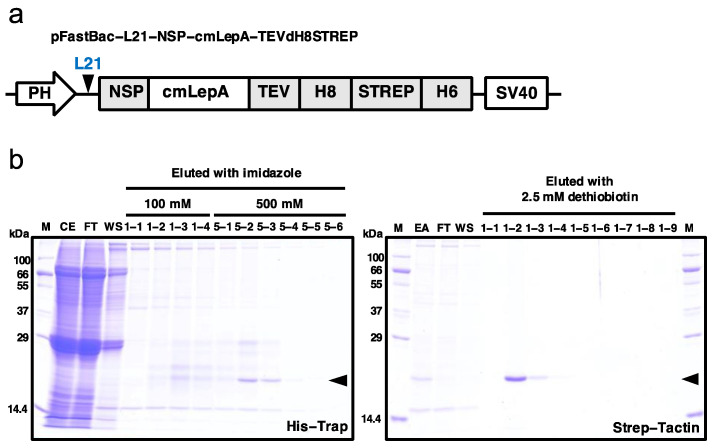
(**a**) Schematic representation of the construct to generate a recombinant baculovirus for leptin expression. The expression of leptin was under the control of the polyhedrin promoter (PH) and followed by an SV40 polyadenylation signal (SV40). L21: leader sequence for enhancing translation efficiency in baculovirus expression system; NSP: native signal peptide; cm-LepA: chub mackerel leptin-a, H8: 8x Histidine tag; STREP: Strep-tag; TEV: tobacco etch virus protease cleavage site. (**b**) Purification of leptin from silkworm pupae infected with the recombinant BmNPV via Nickel (His-Trap, left panel) and Strep-Tactin (right panel) affinity chromatography. The arrow indicates the expression of leptin in silkworm pupae. M: molecular weight markers; CE: crude extract; FT: flow-through; WS: wash fraction; EA: eluted fractions (lane 1-1–5-6, 10 μL applied from 3 mL of each eluted fraction) from His-Trap was concentrated and diluted with PBS (−) and further purified by Strep-Tactin affinity chromatography. The leptin was eluted by 2.5 mM desthiobiotin on the Strep-Tactin (lanes 1-1–1-9, 10 μL applied from 3 mL of each eluted fraction). Protein samples from each step were resolved by 12% SDS-PAGE and visualized using Coomassie Brilliant Blue (CBB) R-250 staining.

**Figure 2 cells-10-03505-f002:**
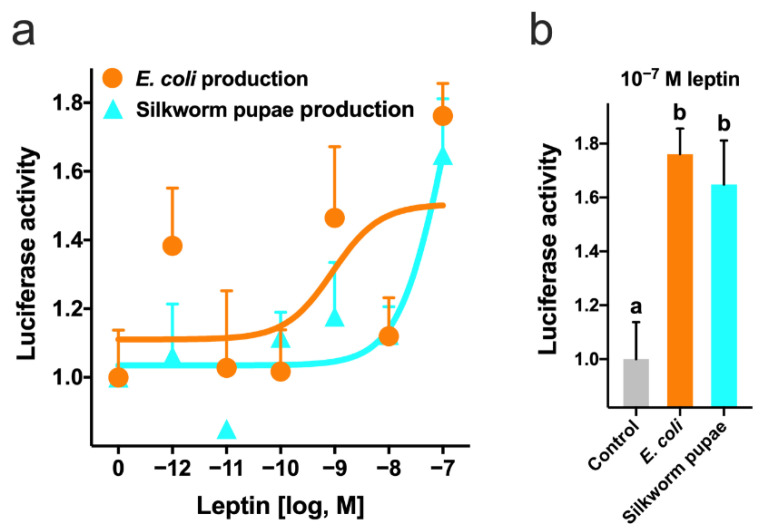
Receptor-binding ability of recombinant leptin produced in *Escherichia coli* and silkworm pupae. (**a**) LepR-transfected cells were treated with graded concentrations of leptin for 3 h. (**b**) Luciferase activity at a ligand concentration of 10^−7^. Data represent the ratio of change in firefly luciferase activity to control *Renilla* luciferase activity. Each point was determined in quadruplicate and is represented as the mean ± SEM; different letters above the bars represent significant differences (*p* < 0.05).

**Figure 3 cells-10-03505-f003:**
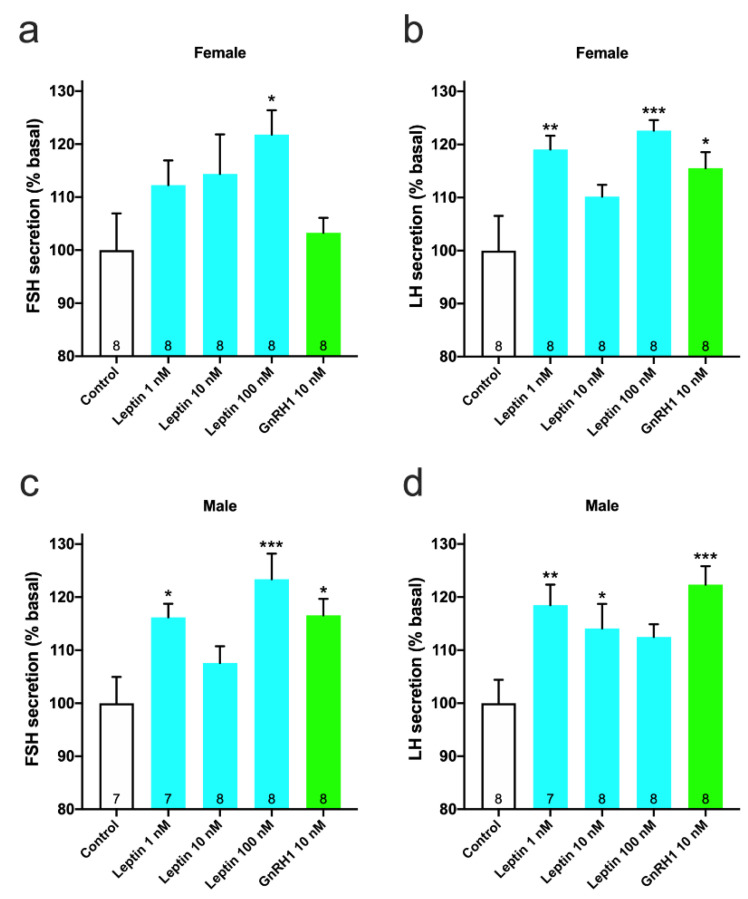
In vitro effect of leptin on GTH secretion from primary pituitary cells. Cells were treated with graded concentrations of leptin for 1 h. (**a**) FSH and (**b**) LH secretion in females and (**c**) FSH and (**d**) LH secretion in males. Data represented the mean ± SEM and were analyzed by one-way ANOVA, followed by a Dunnett’s multiple comparison test (vs. control group). The number at the bottom of the column indicates the number of samples. * *p* < 0.05, ** *p* < 0.01, *** *p* < 0.001.

**Figure 4 cells-10-03505-f004:**
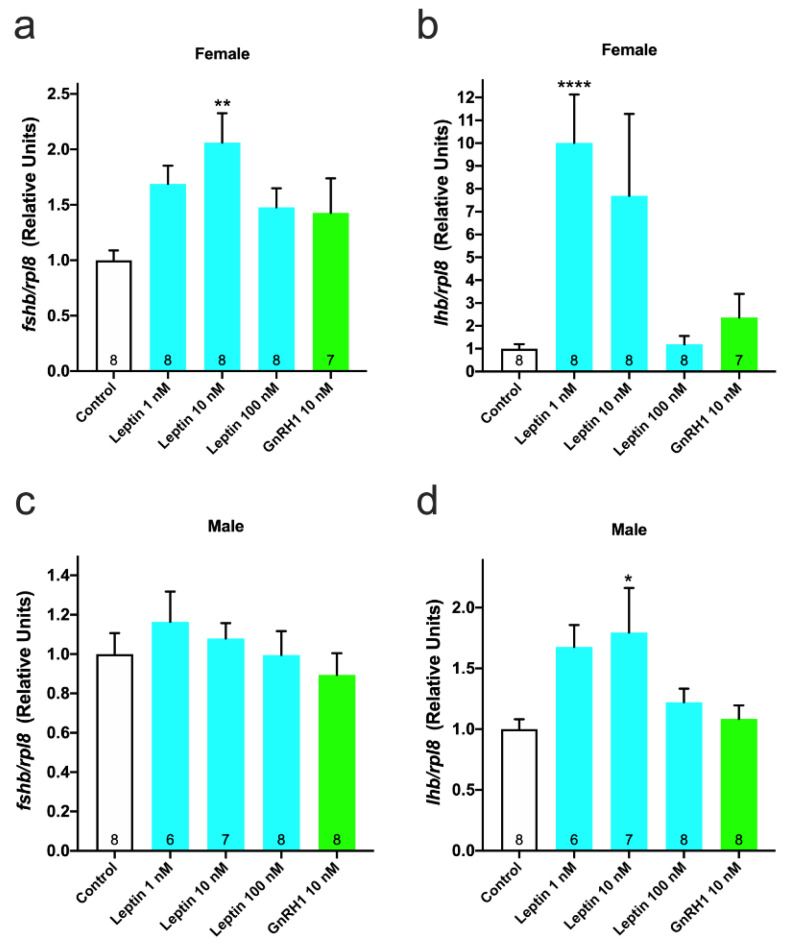
In vitro effect of leptin on GTH gene expression in primary pituitary cells. Cells were treated with graded concentrations of leptin for 1 h. (**a**) *fshb* and (**b**) *lhb* expression in females and (**c**) *fshb* and (**d**) *lhb* expression in males. Data represented the mean ± SEM and were analyzed by one-way ANOVA, followed by a Dunnett’s multiple comparison test (vs. control group). The number at the bottom of the column indicates the number of samples. * *p* < 0.05, ** *p* < 0.01, **** *p* < 0.0001.

**Figure 5 cells-10-03505-f005:**
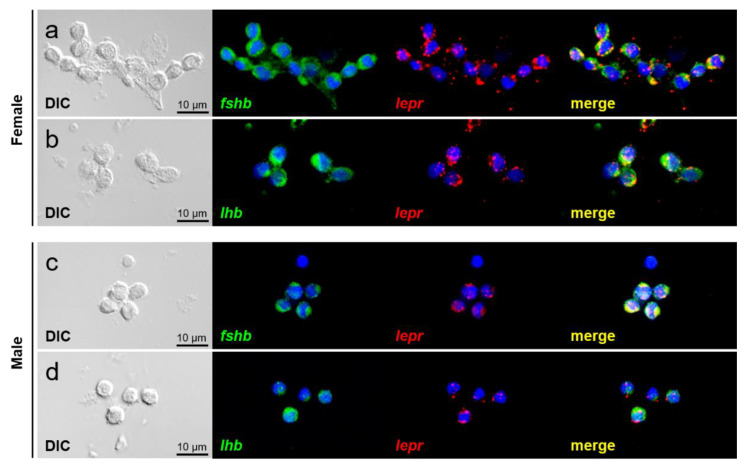
Double-label in situ hybridization in the primary cultured pituitary cells of females (**a**,**b**) and males (**c**,**d**). Cells were visualized by DIC image and DAPI staining of cell nuclei. Green fluorescence indicates *fshb* or *lhb* mRNA expression. Red fluorescence indicates *lepr* mRNA expression. Merged images show that FSH- and LH-producing cells co-expressing *lepr* mRNA. Scale bars = 10 μm.

**Figure 6 cells-10-03505-f006:**
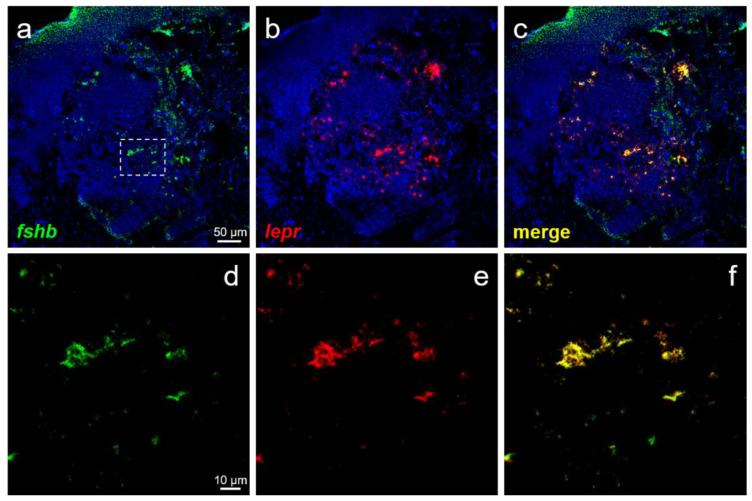
Double-label ISH in the pituitary of females. (**a**) Green fluorescence indicates the *fshb* mRNA-expressing cells. (**b**) Red fluorescence indicates the *lepr* mRNA-expressing cells. (**c**) Merged image of (**a**) and (**b**). (**d**–**f**) Magnified images of FSH-producing cells (the area surrounded by a square in panel **a**) showed that FSH-producing cells co-expressed *lepr* mRNA. (**a**–**c**): scale bars = 50 μm; (**d–f**): scale bars = 10 μm.

**Figure 7 cells-10-03505-f007:**
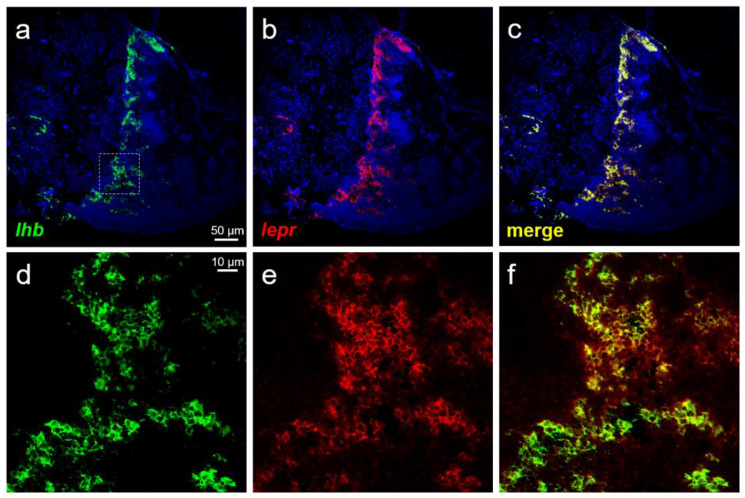
Double-label ISH in the pituitary of females. (**a**) Green fluorescence indicates the *lhb* mRNA-expressing cells. (**b**) Red fluorescence indicates the *lepr* mRNA-expressing cells. (**c**) Merged image of (**a**) and (**b**). (**d**–**f**) Magnified images of LH-producing cells (the area surrounded by a square in panel **a**) showed that LH-producing cells co-expressed *lepr* mRNA. (**a**–**c**): scale bars = 50 μm; (**d**–**f**): scale bars = 10 μm.

**Table 1 cells-10-03505-t001:** Primer sequences used in quantitative real-time PCR.

cDNA	Primer Sequence (5′–3′)	
*fshb*	Fw	TGTGAAGGACAGTGTTACCACAGGG
	Rv	TCATAGGTCCAGTCACCGC
*lhb*	Fw	GAAACAACCATCTGCAGCG
	Rv	AAAAGTCCCGATACGYGCAC
*rpl8*	Fw	CCGCGCTTCAGGAAACTAC
	Rv	TCAATACGACCACCTCCAGC

**Table 2 cells-10-03505-t002:** Primer sequences used in RNA-probe synthesis.

cDNA		Primer Sequence (5′–3′)
*fshb*	Fw	CAAAGCAACAAATCTCTACAGGCG
	Rv	GCAACAAGGAAGGACAATGGG
*lhb*	Fw	GACACCGGCTGAGATTCACTA
	Rv	GTGTGACAACCTTTATTTAGCACAAC
*lepr*	Fw	TGCAGACTATTGAGGCAGAAC
	Rv	CAGACGGGATGGCACCTC

## Data Availability

The data underlying this article will be shared at reasonable request to the corresponding author.

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
