# Peer review of "Leptin Is an Important Endocrine Player That Directly Activates Gonadotropic Cells in Teleost Fish, Chub Mackerel"

_cells, 2021, doi:10.3390/cells10123505_

Round 1

Reviewer 1 Report

Authors described in this manuscript the method to produce chub mackerel leptin using silkworm pupae. They tested the produced leptin, which was as efficient as that obtained using Escherichia coli, on the expression and production of FSH and LH from primary cultured pituitary cells. They also reported the co-expression of leptin receptor with LH or FSH, this has largely been described in different models.

This manuscript is more methodological than mechanistic.

Author Response

Thank you for your helpful comment. This study reports the successful production of recombinant leptin in silk moths. However, the main purpose of this study was to investigate in detail the effect of leptin on GTH secretion and its mechanism of action. To focus more on the functional analysis of leptin, we have now rewritten the background and purpose of the research and reduced the discussion on the leptin production system using the silk moth. However, we have not rewritten the method describing the protein expression by silk moth and its results because we believe that it is necessary to provide this information to the reader. We are confident that this revision focuses on the functional analysis of leptin. It is the first report in fish that shows that leptin is directly involved in the activation of GTH-producing cells, and we believe that it will greatly contribute to the development of reproductive endocrinology research in fish.

We have rewritten the following sections:

Title

Abstract

Introduction: lines 87–97

Discussion: lines 419–433

Reviewer 2 Report

Ohga et al. produced a recombinant leptin in silkworm pupae. Bioactivity of recombinant leptin was confirmed in an in vitro luciferase reporter assay and in primary pituitary cells, where the expression level of FSH and LH beta subunits was determined by QPCR. Localisation of leptin receptor in gonadotrophs was investigated by dual label-in situ hybridisation in dispersed pituitary cells and in pituitary sections.

The aim of the research work was well-defined. The methods employed and data analysis tools were appropriate. The results were presented concisely. The discussion is comprehensive and puts into context the findings of the study. It seems the issue during purification of the recombinant has been resolved and in vivo studies are forthcoming. The authors must be commended for their continuing efforts to advance the understanding of the role of leptin on fish reproduction.

Author Response

Thank you for evaluating our research. As commented by you, we will continue to elucidate the function of leptin in fish reproduction.

Reviewer 3 Report

General comments

This study examines the use of silkworm pupae to produce recombinant chub mackerel leptin vs. E. coli and repeating studies with the new hormone to evaluate the effects of leptin on pituitary gonadotrophs. It appears the silkworm system is better for the elimination of inclusion bodies and folding, however, with the low yield and solubility issues, and lack of receptor saturation, it is unclear if it is truly advantageous over the E. coli expression system. Additionally, the authors also show co-localization of lepr with fshb and lhb in the pituitary, and the LH and FSH synthesis and secretion is regulated by leptin. Understanding leptin’s role in regulating gonadotrophs and reproductive development in fish is important. The data are discussed with respect to the existing literature. Overall, the manuscript is well-written, and for the most part the data are presented clearly. I have a few minor questions and specific comments.

Specific Comments

Line 126: Did the dialysis completely remove the HIS/strep tags from the leptin?

Line 145: State which leptins and at what concentrations they were used in the assay.

Line163: Again, state the hormone concentrations used.

How physiologically relevant are the doses you used in vitro? How did you choose these amounts? While some instances the low doses elicit a response, 100 nM is supraphysiological. Do you know what baseline levels of leptin should be in chub mackerel? Have you tried to measure leptin protein levels in vivo?

Line 175: Define fshb and lhb.

Line 202: Were these pituitary cells harvested from pre-pubescent fish as well? Only females?

Figure 2: The silkworm pupae receptor binding curve does not appear to reach saturation. Why? Perhaps 3 hours was not long enough. Did you run the incubation for longer or run the curve out with higher leptin concentration?

Results: Provide information and tank conditions on where chub mackerel fish samples came from.

Sections 3.2., 3.3 and throughout results section: Include actual data and P-values in the results section, not just in figure legends. For example, where it says “LH secretion was significantly promoted” there should be levels of LH reported with the associated P-value.

Figure 5: It would be helpful to show a female symbol next to figures a/b and male next to c/d to better clarify the differences in the figure panels.

Figures 6/7: Why did the authors only show females? If males were also used, even if there is no co-localization it would be beneficial to show this figure.

Line 370: With solubility issues and low yield, does using silkworm pupae benefits outweigh that of the E. coli system? This should be clarified.

Line 367: Include citation supporting the use of silkworms for recombinant protein production.

Line 378: … “compared to that of the control”

Line 379: cm-rLep produced from E. coli and silkworm both produced bioactive proteins as they both elicited a response from the cells. Were there any notable differences in the responses?

Line 383: Since there appears to be a difference between individuals sampled closer to maturity, the authors should more clearly define the term “pre-puberty” and what physiological or physical markers they are using to distinguish these fish from more “immature” ones. Perhaps future studies could include a developmental puberty time course, sampling at different stages.

Line 416: Leptin appears to have had a greater impact on fshb and lhb expression from female pituitary cells vs. those from males. Secretion from male cells (Figure 3) was significant but expression data show only 10 nM leptin was able to stimulate lhb, and no changes in fshb in male pituitary cells (Figure 4). Further explanation on the different responses should be provided and needs to be addressed here in the discussion.

Line 419: Define “short time”.

Author Response

Specific comments

  • Line 126: Did the dialysis completely remove the HIS/strep tags from the leptin?

Reply

Thank you for your comment. Dialysis was done to replace the buffer. The His and Strep tags were not removed from the recombinant leptin. This information can be found in the original draft (line 248). Since all of these tags are very low-molecular-weight proteins, it is presumed that they do not affect the structure or function of leptin.

  • Line 145: State which leptins and at what concentrations they were used in the assay.

Reply

Thank you for your comment. We have now revised the manuscript according to the comment.

Lines 171–172

  • Line163: Again, state the hormone concentrations used.

Reply

Thank you for your comment. We have now revised the corresponding text.

Line 196-198

  • How physiologically relevant are the doses you used in vitro? How did you choose these amounts? While some instances the low doses elicit a response, 100 nM is supraphysiological. Do you know what baseline levels of leptin should be in chub mackerel? Have you tried to measure leptin protein levels in vivo?

Reply

Thank you for your helpful comment. The leptin concentration used in the experiment was determined by referring to the previous studies that measured blood leptin concentration in other species of fish. The blood leptin concentration in chub mackerel is not known. We are currently developing an ELISA test for measuring blood leptin levels. The concentration of 100 nM may be higher than the physiological concentration. However, there may be slight difference in bioactivity of native leptin and tag-fused recombinant leptin. Therefore, we have not discussed the optimum concentration in this paper.

  • Line 175: Define fshb and lhb.

Reply

Thank you for your comment. We have now revised the manuscript according to the comment.

Line 214-215

  • Line 202: Were these pituitary cells harvested from pre-pubescent fish as well? Only females?

Reply

Pituitaries for ISH analysis in isolated cells were taken from prepubertal males and females. We have now added this information.

Line 243-247

  • Figure 2: The silkworm pupae receptor binding curve does not appear to reach saturation. Why? Perhaps 3 hours was not long enough. Did you run the incubation for longer or run the curve out with higher leptin concentration?

Reply

Thank you for your comment. The incubation time was determined based on the preliminary experiments; where 3 hours period was found to be optimal. Beyond that time, the entire baseline was seen to decrease. We agree with your opinion that it should reach a plateau., We think that it may be necessary to use higher concentrations of leptin for the binding curve to reach saturation. We could not test the leptin produced in silk moths at a concentration higher than 10-7 M because of the relationship between concentration and yield. We would like to investigate the optimum concentration of leptin in an assay in detail in the future.

  • Results: Provide information and tank conditions on where chub mackerel fish samples came from

Reply

We have now added the information about the fish used in the experiments to the method section.

Lines 178–183

We have now furnished the information regarding animal use ethics.

Lines 204–206

  • Sections 3.2., 3.3 and throughout results section: Include actual data and P-values in the results section, not just in figure legends. For example, where it says “LH secretion was significantly promoted” there should be levels of LH reported with the associated P-value.

Reply

Thank you for your comment. We have now revised the manuscript according to the comment.

Please see Section 3.2 and 3.3.

  • Figure 5: It would be helpful to show a female symbol next to figures a/b and male next to c/d to better clarify the differences in the figure panels.

Reply

We have now revised Figure 5 as per the comment.

Please see Figure 5.

  • Figures 6/7: Why did the authors only show females? If males were also used, even if there is no co-localization it would be beneficial to show this figure.

Reply

Thank you for your comment. Frozen samples for making the frozen sections were taken only from the female fish. Therefore, ISH using frozen sections is performed only in females. Co-localization with LepR was confirmed by the ISH using  FSH and LH-producing cells isolated from males., Thus,  we think that leptin's involvement in GTH is common in both males and females.

  • Line 370: With solubility issues and low yield, does using silkworm pupae benefits outweigh that of the E. coli system? This should be clarified.

Reply

Thank you for your comment. It is a characteristic of leptin that it easily aggregates and becomes insoluble, regardless of the expression system. The benefit of using a production system using silk moth is that it is easier to scale up than the E. coli expression system. This was mentioned in the original manuscript in lines 362–364.

In the revised manuscript, we have included the technical part of protein production by silk moth is a simplified language in response to the comment by other reviewers.

  • Line 367: Include citation supporting the use of silkworms for recombinant protein production.

Reply

Thank you for your comment. Since the advantages of producing recombinant proteins using silk moths mentioned in the manuscript are common opinion, it is difficult to cite a specific academic paper for it.

Line 378: … “compared to that of the control”

Reply

Thank you for your comment. We have now revised the manuscript as per your comment.

Line 457.

  • Line 379: cm-rLep produced from E. coli and silkworm both produced bioactive proteins as they both elicited a response from the cells. Were there any notable differences in the responses?

Reply

Since we have not compared the bioactivity of silkworm-produced leptin and E. coli-produced leptin using the same pituitary sample, we are not able to compare their bioactivities. However, we did not notice a significant difference.

  • Line 383: Since there appears to be a difference between individuals sampled closer to maturity, the authors should more clearly define the term “pre-puberty” and what physiological or physical markers they are using to distinguish these fish from more “immature” ones. Perhaps future studies could include a developmental puberty time course, sampling at different stages.

Reply

Thank you for your important comment. Since no specific physical markers are used to distinguish between different stages of maturation, it is not possible to classify immature states into individual stages at present time. Therefore, we have now changed the expression "more immature individual" to "more juvenile individual". In the future, as per your advice, we would like to use appropriate markers to distinguish between different reproductive stages.

Line 464-467

  • Line 416: Leptin appears to have had a greater impact on fshb and lhb expression from female pituitary cells vs. those from males. Secretion from male cells (Figure 3) was significant but expression data show only 10 nM leptin was able to stimulate lhb, and no changes in fshb in male pituitary cells (Figure 4). Further explanation on the different responses should be provided and needs to be addressed here in the discussion.

Reply

Thank you for your helpful comment. We have now discussed this in the revised manuscript.

Lines 506–512

  • Line 419: Define “short time”.

Reply

We have now revised the manuscript to clearly define “short time.”

Lines 501